# iGRPO: Fast Online RL for Flow Matching Model with Instant Reward

**Sucheng Ren** [1 2]  **Chen Chen** [2]  **Zhenbang Wang** [2]  **Liangchen Song** [2]  **Xiangxin Zhu** [2]  **Yinfei Yang** [2]  **Jiasen Lu** [2]

## Abstract

Conventional practice assumes that online reinforcement learning for flow-matching models requires sampling full denoising trajectories to compute rewards. This assumption underlies methods such as Group Relative Policy Optimization (GRPO), where the policy must traverse the entire reverse process before receiving a delayed, trajectory-level reward. We observe, however, that while such terminal rewards provide feedback, they are neither necessary nor optimal for effective learning. In this work, we introduce iGRPO (Instant-reward GRPO), which replaces GRPO's full-trajectory rollouts with a single-step mapping that assigns rewards instantly at each denoising step. Because the flow matching model behaves differently across timesteps, our step-local instant rewards which are inherently time-dependent, overcome prior approaches that rely on a single, time-independent terminal reward. By evaluating each action locally rather than relying on a final terminal score, iGRPO eliminates the need for multi-step SDE rollouts and offers more precise credit assignment. Across standard benchmarks, iGRPO converges 10.2× faster than FlowGRPO while achieving higher final alignment quality. We hope this work motivates more efficient and scalable online RL methods for flow-matching generative models.

## 1. Introduction

Flow-matching models (Lipman et al., 2022; Esser et al., 2024; Labs, 2024) have recently emerged as a dominant approach for high-fidelity image generation. Their strong theoretical grounding and efficient deterministic samplers enable high-quality synthesis across diverse domains. Despite this progress, these models still struggle with tasks that

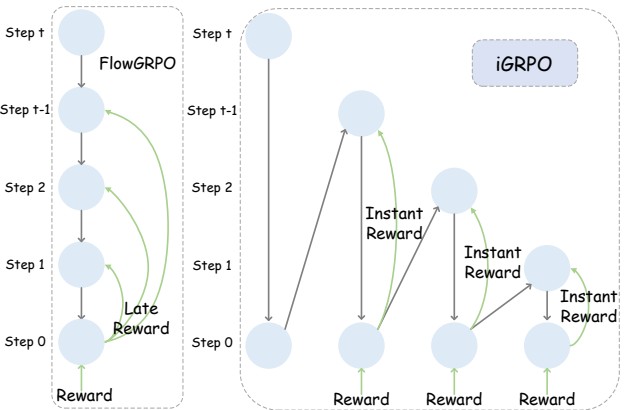

*Figure 1.* Comparison between FlowGRPO (left) with late reward and our iGRPO (right) with instant reward. our iGRPO avoid sampling the whole trajatory and the instant reward better reflect the timestep-related criticality of each denoise stage.

require precise compositional control – such as counting, spatial relations, attribute binding – and robust text rendering. These limitations have motivated growing interest in aligning flow models with task-specific objectives.

Online reinforcement learning (RL) offers a natural way for this alignment. Recent frameworks such as Group Relative Policy Optimization (GRPO) (Shao et al., 2024) optimize generative policies directly toward reward functions, and FlowGRPO (Liu et al., 2025) extends this idea to flow-matching models by injecting stochasticity through an SDE formulation. Given the continuous nature of flow trajectories, this line of work has led to a widely held belief: applying RL to flow models requires sampling full denoising trajectories and assigning rewards only at the terminal state.

In this work, we ask a simple question: *Is it necessary for RL in flow-matching models to rely on full trajectories and terminal rewards?* We observe that the two defining elements of FlowGRPO – (1) traversing the complete denoising trajectory and (2) evaluating reward only at the final image – are not conceptually required. The denoising steps exist to reach the terminal sample, but the learning signal itself depends solely on the evaluation of generated data. Likewise, while terminal rewards are convenient, they obscure the heterogeneous importance of different denoising stages:

[1]Johns Hopkins University [2]Apple. Correspondence to: Jiasen <jiasen.lu@apple.com>.

*Proceedings of the 43rd International Conference on Machine Learning*, Seoul, South Korea. PMLR 306, 2026. Copyright 2026 by the author(s).

early timesteps explore broad regions of the latent space, whereas later ones make fine-grained corrections.

This leads to two key observations. First, full denoising trajectories are not inherently needed for RL. If a model can map noisy states directly to clean data, the intermediate path becomes optional. Second, terminal rewards are not exactly aligned with the denoising timesteps. A single delayed score cannot adequately credit actions taken across timesteps of differing significance.

With these observations, we propose iGRPO (Instant-reward GRPO), a reinforcement-learning framework based on single-step rollout and instant reward assignment rather than full-trajectory sampling, as illustrated in Figure 1. Inspired by consistency-style models (Song et al., 2023), iGRPO performs a single-step mapping from noise (or noisy data) directly to the data space, then re-noises back to intermediate states to retain sufficient stochasticity for exploration. This design removes the dependency on multi-step rollouts and replaces sparse terminal rewards with instant feedback that reflects timestep-dependent criticality.

Our approach yields significant practical benefits. As shown in Figure 2, FlowGRPO (Liu et al., 2025) converges at around 10k iterations, whereas iGRPO reaches a comparable reward in only 1k iterations and ultimately attains a higher final reward. These results demonstrate that tightly coupling RL to full denoising trajectories –incurring extremely long sampling times and delayed rewards that fail to reflect intermediate states – is not an inherent requirement of flow-matching models. We also show that our instant reward is time-dependent: as illustrated in Figure 4, it induces higher exploration at early timesteps and lower exploration at later timesteps, aligning with the characteristic behavior of flow-matching inference. We hope this work broadens the design space of RL-based alignment for continuous generative models and motivates further exploration of step-local reward strategies. Leveraging both shortened rollouts and time-dependent instant rewards, and using CLIPScore as the reward, iGRPO outperforms FlowGRPO by 10.8% on GenEval with only 1k iterations – just 10% of FlowGRPO's training iterations.

## 2. Related Work

**Flow Matching and Diffusion.** Diffusion models (Ho et al., 2020; Song & Ermon, 2020; Song et al., 2021) have become a standard backbone for high-fidelity generative modeling by learning to invert a progressive noising process. Training objectives include denoising or score matching, and synthesis is performed by integrating a learned reverse-time process, either stochastic (SDE) or deterministic via the probability flow ODE. Flow matching (Lipman et al., 2022; 2023; Albergo & Vanden-Eijnden, 2023) generalizes this

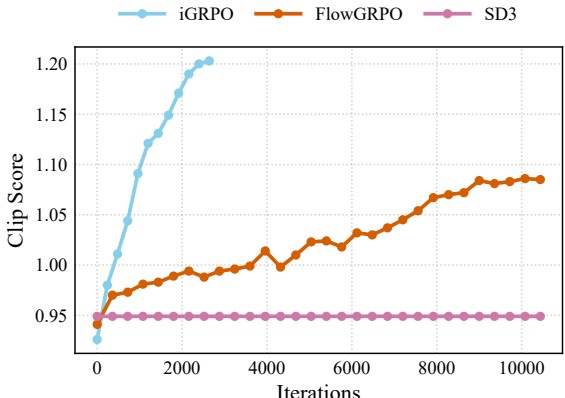

*Figure 2.* ClipScore over training iterations. Under the same reward (ClipScore), iGRPO at 1k iterations achieve the same reward as FlowGRPO at 10k iterations. iGRPO attend a better convergence at the end.

paradigm by learning a time-conditioned velocity field that deterministically transports a simple prior to the data distribution, unifying ideas from diffusion and normalizing flows while naturally supporting adaptive-step ODE samplers.

Recent large-scale applications, such as SD3 (Esser et al., 2024) and FLUX (Labs, 2024), demonstrate that flow-matching models can serve as foundation models for multi-modal generation. Despite high fidelity, inference typically requires many function evaluations, which slows sampling and complicates online RL fine-tuning due to trajectory-level, delayed rewards. Consistency Models (CMs) (Song et al., 2023) address this by learning one-step mappings that maintain quality across noise levels, inspiring our use of single-step mappings within the RL loop to expose dense, step-local rewards. At inference, we retain standard ODE samplers, preserving generation fidelity.

**Reinforcement learning for text-to-image alignment.** Beyond supervised pretraining, reinforcement learning (RL) has proven effective for aligning generative models to human preferences (Ouyang et al., 2022; Schulman et al., 2017; Rafailov et al., 2023). Recent work introduces Group Relative Policy Optimization (GRPO) (Shao et al., 2024), which stabilizes on-policy updates by normalizing rewards within groups, improving robustness under noisy preference signals.

For text to image generation, RL (Liu et al., 2025; Xue et al., 2025) align generative image models with human or programmatic feedback, targeting compositional correctness, OCR fidelity, aesthetics, or safety. Previous RL-based alignment for T2I clusters into three practical strands. Direct fine-tuning with differentiable rewardstreat alignment as a differentiable objective defined on model samples and backpropagate reward gradients through (an approximation of) the sampler. ImageReward (Xu et al., 2023) introduced

a learned preference model and an associated direct tuning procedure. DRaFT (Clark et al., 2023) shows strong gains without on-policy rollouts. AlignProp (Prabhudesai et al., 2023) likewise propagates reward gradients end-to-end for large-scale diffusion alignment. RENO (Eyring et al., 2024) optimizes the initial noise for one-step generators with reward-guided updates. Another way is DPO-style preference (Rafailov et al., 2023) optimization which replaces on-policy RL with preference-risk minimization on paired or implicit comparisons, led by Diffusion-DPO (Wallace et al., 2024). Extensions include DSPO, which operates in diffusion score space (Zhu et al., 2025), and related KL-regularized preference formulations for diffusion fine-tuning (Fan et al., 2023). These approaches leverage large offline corpora of preferences and avoid sampling variance during training. Besides, PPO-style policy gradients (Schulman et al., 2017) keeps an explicit MDP over denoising and optimize expected reward online. DDPO (Black et al., 2023) is the canonical approach and demonstrates prompt-image alignment improvements using learned or heuristic rewards.

**FlowGRPO, DanceGRPO, and iGRPO.** Flow-GRPO (Liu et al., 2025) applies GRPO to flow models by converting the ODE sampler into an SDE and reducing the number of optimized denoising steps. Nevertheless, training still relies on multi-step rollouts and group-relative rewards aggregated at the end of the trajectory. DanceGRPO (Xue et al., 2025) further stabilizes GRPO across diffusion and rectified-flow models, multiple tasks, and diverse reward types.

In contrast, Our iGRPO performs a single-step rollout and re-noises the generated sample to inject controlled stochasticity for exploration. Crucially, it assigns *instant, timestep-dependent rewards*, rather than terminal, trajectory-level rewards. This dense, step-local feedback yields sharper credit assignment and substantially lower sampling cost, addressing the inefficiencies of prior methods while preserving the flow-matching backbone.

## 3. Method

**Overview.** We propose iGRPO (Instant-reward GRPO) which replaces trajectory-wise ODE integration to a single step mapping and supplies instant, step-local rewards, while preserving stochastic exploration via a lightweight re-noising mechanism. iGPRO retains the exploration benefits of stochastic sampling used in prior flow-based RL, while removing the requirement to sample entire reverse trajectories, thereby reducing variance and wall-clock cost without sacrificing image fidelity. We begin by revisiting flow matching and flowgrpo.

### 3.1. Preliminaries

**Flow matching.** Let $p_{data}(x)$ denote the data distribution and $p_0(x)$ a simple prior (e.g., $\mathcal{N}(0, I)$). Flow matching trains a time–conditioned velocity field $v_\theta(x, t)$ whose deterministic dynamics

$$\frac{dx_t}{dt} = v_\theta(x_t, t, c), \qquad t \in [0, 1], \tag{1}$$

transport $x_1 \sim p_1(\cdot) \equiv p_0$ to $x_0 \sim p_{data}$. Training is cast as supervised regression to a target field $v^\star(x, t)$ sampled from an interpolation between a data sample $x_{data}$ and a prior sample $x_0$ with straight paths $\psi_t(x_0, x_{data}) = (1 - t)\, x_0 + t\, x_{data}$ and constant target velocity $v^\star(x_t, t) = x_{data} - x_0$. At inference, one integrates (1) from $t = 1$ to $t = 0$ (often with a small number of ODE steps and optional higher-order solvers), yielding a sample $\hat{x}_0$ conditioned on text prompt $c$ via a conditioning mechanism in $v_\theta(\,\cdot\,, t \mid c)$.

**FlowGRPO.** FlowGRPO introduce GRPO into flow matching. Given a prompt $c$, generation induces an MDP where the state encodes the current noisy point and time, $s_t = (x_t, t, c)$, and the policy is the conditional generator defined by the flow matching model (e.g., an ODE step or an SDE step). A reward model $R(\hat{x}_0, c)$ evaluates terminal samples (at step 0). Group Relative Policy Optimization (GRPO) stabilizes on-policy learning by normalizing rewards within a group of $G$ samples for the same prompt. If $\{\hat{x}_0^{(i)}\}_{i=1}^G$ are group members with rewards $r^{(i)} = R(\hat{x}_0^{(i)}, c)$, GRPO forms standardized advantages

$$\begin{aligned}
\hat{A}^{(i)} &= \frac{r^{(i)} - \mu_r}{\sigma_r}, \\
\mu_r &= \frac{1}{G} \sum_j r^{(j)}, \\
\sigma_r &= \sqrt{\frac{1}{G} \sum_j (r^{(j)} - \mu_r)^2 + \varepsilon}.
\end{aligned} \tag{2}$$

Let $\pi_\theta(a \mid s)$ denote the policy induced by the flow matching model over its stochastic transition (SDE in practice). The GRPO objective adopts PPO-style clipping over the likelihood ratio $\rho^{(i)}(\theta) = \frac{\pi_\theta(a^{(i)} \mid s^{(i)})}{\pi_{\theta_{old}}(a^{(i)} \mid s^{(i)})}$:

$$\begin{aligned}
\mathcal{J}(\theta) = {}& \mathbb{E}_{t \sim p(t)} \left[ \frac{1}{G} \sum_{i=1}^G \ell_t^{(i)}(\theta) \right] \\
& - \beta\, \mathbb{E}_{t \sim p(t)} \left[ D_{KL} \big( \pi_\theta(\cdot \mid s_t) \,\big\|\, \pi_{ref}(\cdot \mid s_t) \big) \right],
\end{aligned} \tag{3}$$

with an optional KL regularizer to a reference policy $\pi_{ref}$. Where

$$\ell_t^{(i)}(\theta) = \min\!\left( \rho_t^{(i)}(\theta)\, \hat{A}_t^{(i)},\ \text{clip}\big(\rho_t^{(i)}(\theta), 1-\epsilon, 1+\epsilon\big)\, \hat{A}_t^{(i)} \right) \tag{4}$$

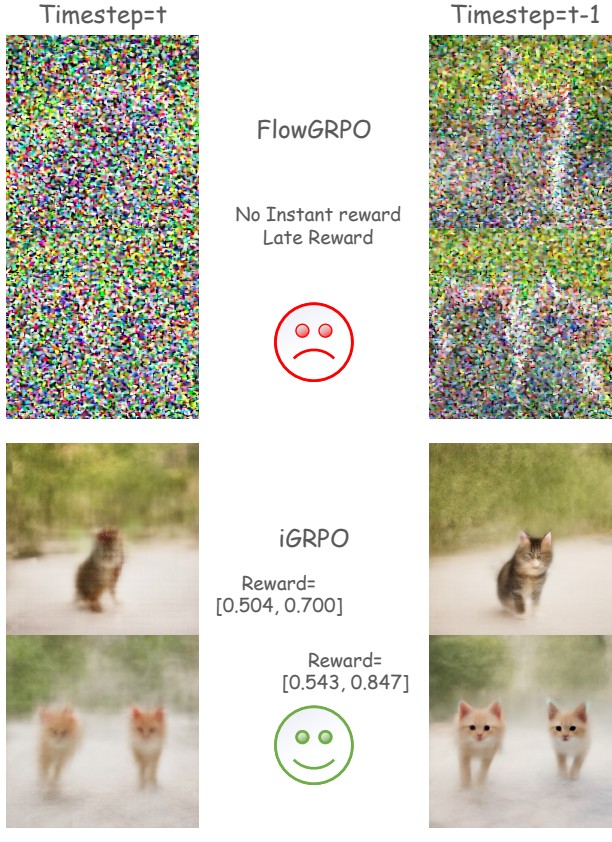

Timestep=t    Timestep=t-1

FlowGRPO

No Instant reward
Late Reward

iGRPO

Reward=
[0.504, 0.700]

Reward=
[0.543, 0.847]

*Figure 3.* Late vs. instant reward. In FlowGRPO, the policy must generate the entire denoising trajectory before receiving a single terminal reward, therefore, intermediate states are too noisy and blurry to provide meaningful instant reward. In contrast, iGRPO obtains reward immediately on each one-step prediction, enabling dense, step-local reward assignment.

Although FlowGRPO can reduce the step count (e.g., from 40 to 10) via denoising reduction, it still samples full reverse trajectories and only receives reward at the terminal time step, which dilutes reward assignment for intermediate actions.

### 3.2. iGRPO

We retain a velocity parameterization. Given a reverse step size $\Delta t > 0$ (from $t$ to $t - \Delta t$), define the mean:

$$\mu_\theta(x_t, t, c; \Delta t) \;=\; (1 - (t - \Delta t))(x_t - t v_\theta(x_t, t, c)), \quad (5)$$

During the rollout we re-noise $\hat{x}_0$ to the natural rectified-flow marginal at noise level $t - \Delta t$ with a fresh Gaussian sample,

$$x_{t-\Delta t} \;=\; \mu_\theta(x_t, t, c; \Delta t) \;+\; (t - \Delta t)\,\epsilon, \quad (6)$$

and a small exploration covariance:

$$\Sigma_\theta(t, \Delta t) \;=\; \sigma_t^2\,\Delta t\,I, \quad (7)$$

where

$$\sigma_t \;=\; a\,\sqrt{\tfrac{t}{1-t}}, \quad a \in (0,1) \quad (8)$$

Our policy kernel is then the Gaussian

$$p_\theta(x_{t-\Delta t} \mid x_t, c) \;=\; \mathcal{N}\Big(x_{t-\Delta t}\,;\,\mu_\theta(x_t, t, c; \Delta t),\,\Sigma_\theta(t, \Delta t)\Big). \quad (9)$$

This kernel is a drop-in replacement for the SDE transition used in FlowGRPO: it preserves a closed-form likelihood, supports GRPO ratios, and admits an analytic (per-step) KL to a reference kernel. The new log-probability is:

$$\begin{aligned}
\log \pi_\theta(a_t \mid s_t) = &- \tfrac{d}{2}\log(2\pi), \\
&- \tfrac{1}{2}\log\big|\Sigma_\theta(t, \Delta t)\big|, \\
&- \tfrac{1}{2}\big\|x_{t-\Delta t} - \mu_\theta(x_t, t, c; \Delta t)\big\|_{\Sigma_\theta^{-1}}^2,
\end{aligned} \quad (10)$$

**Instant Reward.** A practical difficulty in on-policy RL for generative flows is that the reward $R(\cdot, c)$ is naturally defined only on terminal (clean) samples at $t=0$, whereas the policy $\pi_\theta$ acts on intermediate states $(x_t, t)$. In our iGRPO, after sampling $x_{t-\Delta t} \sim p_\theta(\cdot \mid s_t)$, we form a post-action terminal by a stop-gradient collapse from $t - \Delta t$ to $0$:

$$\hat{x}_0^{\text{post}} \;=\; x_{t-\Delta t} - (t - \Delta t)\,v_{\tilde\theta}(x_{t-\Delta t},\, t - \Delta t, c). \quad (11)$$

where gradients are stopped through $\tilde\theta$. We then evaluate the reward immediately:

$$r_t \;=\; R\big(\hat{x}_0^{\text{post}},\, c\big). \quad (12)$$

and attribute $r_t$ to the action that produced $x_{t-\Delta t}$ via the GRPO surrogate in Eq. (3). This yields a strictly on-policy, single-step reward assignment for terminal-only reward models, avoiding full stochastic rollouts.

**Anlysis and Discussion.** FlowGRPO mandates sampling the entire reverse trajectory; stochasticity injected early is often attenuated by subsequent integration, and reward arrives only at $t=0$, diluting reward assignment. In contrast, iGRPO evaluates each action immediately via collapse-and-score, so every decision receives direct feedback on its consequence, yielding more effective exploration and sharper reward assignment. iGRPO also samples reverse time $t$ and step size $\Delta t$, covering multiple noise and step-size regimes and expanding the explored neighborhood around the reference path.

As shown in Fig. 4, we compare the per-step reward variance under three sampling strategies: ODE, SDE, and ours. Because ODE sampling is deterministic, its reward variance remains zero at all steps. Under SDE with a terminal reward, the same reward is attributed to every step, so the

*Table 1.* Evaluation on Geneval with ClipScore as reward. Our iGPRO makes significant improvements over SD3-M under identical architecture and outperform FlowGRPO with 10% training iterations.

| Model | Overall | Single Obj. | Two Obj. | Counting | Colors | Position | Attr. Binding |
|---|---|---|---|---|---|---|---|
| LDM (Rombach et al., 2022) | 0.37 | 0.92 | 0.29 | 0.23 | 0.70 | 0.02 | 0.05 |
| SD1.5 (Rombach et al., 2022) | 0.43 | 0.97 | 0.38 | 0.35 | 0.76 | 0.04 | 0.06 |
| DALLE-3 (Betker et al., 2023) | 0.67 | 0.96 | 0.87 | 0.47 | 0.83 | 0.43 | 0.45 |
| SD3-M (Esser et al., 2024) | 0.63 | 0.98 | 0.78 | 0.50 | 0.81 | 0.24 | 0.52 |
| SD3-L (Esser et al., 2024) | 0.71 | 0.98 | 0.89 | 0.73 | 0.83 | 0.34 | 0.47 |
| FLUX.1-Dev (Labs, 2024) | 0.66 | 0.98 | 0.81 | 0.74 | 0.79 | 0.22 | 0.45 |
| Show-o (Xie et al., 2024) | 0.53 | 0.95 | 0.52 | 0.49 | 0.82 | 0.11 | 0.28 |
| Emu3 (Wang et al., 2024) | 0.54 | 0.98 | 0.71 | 0.34 | 0.81 | 0.17 | 0.21 |
| JanusFlow (Ma et al., 2025) | 0.63 | 0.97 | 0.59 | 0.45 | 0.83 | 0.53 | 0.42 |
| SD3-M+FlowGRPO (10k iter) (Liu et al., 2025) | 0.74 | 0.99 | 0.89 | 0.77 | 0.84 | 0.32 | 0.64 |
| SD3-M+iGRPO(Ours, 1k iter) | 0.82 | 1.00 | 0.96 | 0.80 | 0.85 | 0.49 | 0.70 |

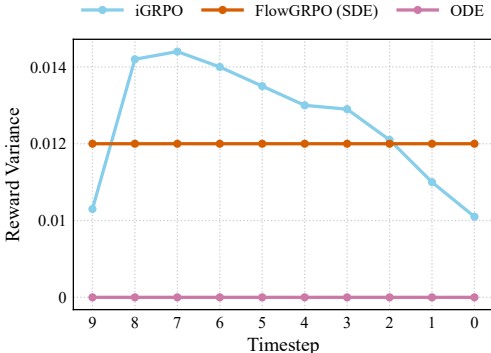

*Figure 4.* Reward variance over timesteps. Under deterministic ODE sampling, intermediate states are fixed, so there is effectively no per-timestep reward variance. FlowGRPO introduces stochasticity via an SDE but still assigns a single terminal reward, yielding a flat reward signal shared by all timesteps. In contrast, iGRPO produces timestep-dependent instant rewards, capturing the heterogeneous variance and criticality of different denoising phases.

per-step reward variance is constant across time. However, in flow matching models different time steps tend to generate different frequency components of the image (Yang et al., 2023; Park et al., 2023; Huang et al., 2024), raising the question of whether a step-independent reward variance is appropriate. For our iGRPO, the reward variance exhibits a U-shaped pattern: it starts relatively high, decreases in the early steps, and rises again in the later steps. This aligns with the coarse-to-fine generation behavior of flow matching – early steps primarily synthesize low-frequency structure that is comparatively easy and consistent (lower variance), whereas later steps focus on high-frequency details that are harder to generate reliably, leading to increased variance.

## 4. Experiment

**Implementation Details.** We consider three configurations: (1) Following FlowGRPO, we take CLIPScore (Radford et al., 2021; Hessel et al., 2021) as the reward model.

We conduct experiments on Pick-a-Pic (Kirstain et al., 2023) dataset. We take stable diffusion (SD3.5-M) (Esser et al., 2024; Yin et al., 2024) as base model. (2) Following Flow-GRPO, we conduct experiments on Geneval (Ghosh et al., 2023) dataset. and take Geneval as the reward. We keep the rest settings the same as (1). (3) Following DanceGRPO, we take Flux (Labs, 2024) as base model and a hybrid reward model combining HPS-v2.1 (Wu et al., 2023) and CLIP-Score (Radford et al., 2021). The training data is HPD (Wu et al., 2023).

All experiments are conducted on 32 H100 GPUs with a batch size of 3 and a maximum of 2500 iterations. We use AdamW (Loshchilov & Hutter, 2017) as the optimizer with a learning rate of 1e-5 and a weight decay coefficient of 0.0001. For GRPO, SD3 model generate 24 samples per prompt while Flux model generate 12 images per prompt.

**Evaluation Metric.** We take five evaluation metric including Geneval (Ghosh et al., 2023), HPS-v2.1 (Wu et al., 2023), CLIPScore (Radford et al., 2021), PickScore (Kirstain et al., 2023), ImageReward (Xu et al., 2023), and Unified Reward (Wang et al., 2025). Geneval is a rule-based benchmark that tests compositional grounding (counts, colors, spatial relations) using programmatic checks on images generated from templated prompts. HPS-v2.1 is Human Preference Score trained on large-scale pairwise comparisons of model outputs. It predicts which image humans would prefer for a given prompt. CLIPScore is the cosine similarity between CLIP image and text embeddings; measuring text–image alignment. PickScore is a learned preference model distilled from the Pick-a-Pic dataset. It estimates human-likeness and prompt faithfulness from single images relative to their prompts. ImageReward is A reward model trained on human comparisons to capture aesthetic quality, semantic alignment, and instruction following. Unified Reward is a single scalar that combines multiple signals (e.g., alignment, aesthetics, preference models) via normalization and learned/weighted aggregation to better

*Table 2.* Evaluation on Geneval with Geneval as reward.

| Model | Overall | Single Obj. | Two Obj. | Counting | Colors | Position | Attr. Binding |
|---|---|---|---|---|---|---|---|
| DALLE-3 (Betker et al., 2023) | 0.67 | 0.96 | 0.87 | 0.47 | 0.83 | 0.43 | 0.45 |
| SD3-M (Rombach et al., 2022) | 0.63 | 0.98 | 0.78 | 0.50 | 0.81 | 0.24 | 0.52 |
| SD3-L (Rombach et al., 2022) | 0.71 | 0.98 | 0.89 | 0.73 | 0.83 | 0.34 | 0.47 |
| FLUX.1-Dev (Labs, 2024) | 0.66 | 0.98 | 0.81 | 0.74 | 0.79 | 0.22 | 0.45 |
| GPT-4o (Hurst et al., 2024) | 0.84 | 0.99 | 0.92 | 0.85 | 0.92 | 0.75 | 0.61 |
| SD3-M+FlowGRPO (1.5k iter) | 0.78 | 0.98 | 0.93 | 0.81 | 0.83 | 0.43 | 0.68 |
| SD3-M+FlowGRPO (8k iter) | 0.95 | 1.00 | 0.99 | 0.95 | 0.92 | 0.99 | 0.86 |
| SD3-M+iGRPO (Ours, 1.5k iter) | 0.96 | 1.00 | 0.98 | 0.96 | 0.94 | 0.97 | 0.92 |

*Table 3.* Evaluation on in domain reward (HPS-v2.1 and CLIPScore) and out-of domain reward (Pick, Score ImageReward and Unified Reward)

| Method | In Domain | | Out-of-Domain | | |
|---|---|---|---|---|---|
| | HPS-v2.1 | CLIPScore | PickScore | ImageReward | Unified Reward |
| SD-XL (Rombach et al., 2022) | 0.280 | 0.287 | 0.224 | 0.76 | 2.931 |
| SD3.5-L (Esser et al., 2024) | 0.288 | 0.289 | 0.228 | 0.960 | 3.253 |
| FLUX.1-Dev (Labs, 2024) | 0.313 | 0.388 | 0.227 | 1.088 | 3.370 |
| DanceGRPO (Xue et al., 2025) | 0.346 | 0.400 | 0.228 | 1.314 | 3.377 |
| iGRPO | 0.351 | 0.418 | 0.235 | 1.441 | 3.521 |

approximate overall human preference..

## 4.1. Quantitative Evaluation

As shown in Table 1, under the same reward model (CLIP-Score), our iGRPO variant trained for only 1k iterations achieves the best overall Geneval score with CLIP reward (0.82), outperforming both base models (SD3-M (Esser et al., 2024): 0.63, SD3-L (Esser et al., 2024): 0.71). Under identical architecture of SD3-M, our method yields consistent gains across key compositional categories: +0.18 on Two Obj. (0.96 vs. 0.78), +0.30 on Counting (0.80 vs. 0.50), +0.25 on Position (0.49 vs. 0.24), and +0.18 on Attribute Binding (0.70 vs. 0.52). Compared to FlowGRPO trained with 10k iteration, our approach trained with 1k iteration improves Overall by +0.08 (0.82 vs. 0.74), with particularly large margins on position (+0.17)

When the reward model goes to Geneval, As shown in Table 2, our method reaches an Overall score of 0.96 with only 1.5k iterations, surpassing FlowGRPO trained for 8k iterations (0.95). It substantially outperforms both base models and short-horizon baselines: vs. SD3-M, Overall improves by +0.32 (0.96 vs. 0.63). Compared with state-of-the-art multimodal generative model, GPT-4o, our iGPRO makes 0.12 improvements.

When the reward model goes to HPS and CLIPScore, we report in-domain metrics (HPS-v2.1, CLIPScore) and out-of-domain metrics (PickScore, ImageReward, Unified Re-

ward). As shown in Table 3, iGRPO improves over both the base FLUX sampler and DanceGRPO across all metrics. Versus FLUX, iGRPO raises HPS from 0.313 to 0.351 (+0.038; +12.1%) and CLIPScore from 0.388 to 0.418 (+0.030; +7.7%). Crucially, out-of-domain generalization also strengthens: PickScore 0.227 to 0.230 (+0.003; +1.3%), ImageReward 1.088 to 1.441 (+0.353; +32.4%), and Unified Reward 3.370 to 3.521 (+0.151; +4.5%). Relative to DanceGRPO, iGRPO still yields consistent gains in HPS (+1.4%), CLIPScore (+4.5%), PickScore (+0.9%), ImageReward (+9.7%), Unified Reward (+4.3%). These improvements indicate that instant, step-local rewards do not merely overfit the in-domain targets, but also transfer to diverse evaluators, aligning the model more robustly with compositional and textual fidelity.

## 4.2. Qualitative Evaluation

As shown in Figure 5, we compare our iGRPO with Flux and FlowGRPO. Our iGRPO achieves better text–image alignment over Flux and FlowGRPO . Additional visualizations are provided in Figure 6, further demonstrating that iGRPO generates high-quality images that align well with human preferences. Prompts can be found in supplementary materials.

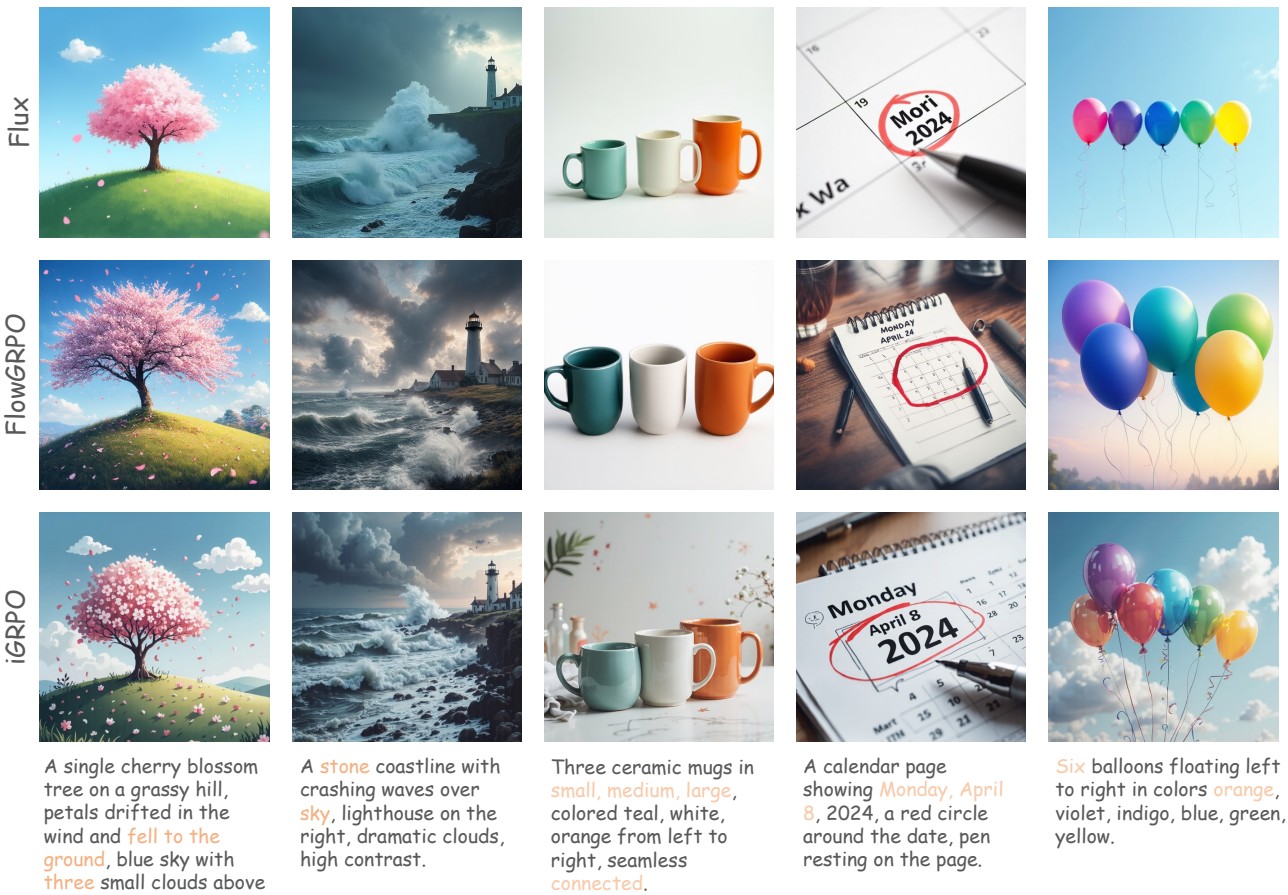

*Figure 5.* Qualitative comparison. Compared with Flux and FlowGRPO, our iGRPO produces sharper, more faithful images with improved counting, spatial relations, attribute binding, and text rendering, leading to stronger text–image alignment.

## 4.3. Ablation Study

**Effectiveness of one step rollout.** Compared with instant reward, our design is also compatible with terminal reward. We compare three reward placements. Terminal (reward is calculate only at $t=0$), Instant (single-step collapse), and a Hybrid mixture (instant and a light terminal reward). As shown in Table 4, iGRPO with terminal reward keeps multi-step rollouts as FlowGRPO can also cut training cost from 2.1k GPU-hours (FlowGRPO) to 860 hours ( 2.4× faster,) while improving Geneval from 0.74 to 0.78 due to faster convergence speed. iGRPO with Hybrid reward further reduces cost to 490 hours (additional 43% speedup) and preserves the 0.80 Geneval, indicating that some instant rewards boost the performance. iGRPO with Instant is the fastest at 205 hours ( 10.2× faster than FlowGRPO), suggesting that purely step-local reward assignment achieve fastest convergence and the time-dependent instant reward boost final performance.

**Reference Policy.** Because iGRPO induces higher action entropy and therefore explores a wider neighborhood around

*Table 4.* Comparison of terminal, hybrid, and instant reward schemes in terms. iGRPO variants reduce training time compared to FlowGRPO while improving alignment performance.

| Method | Reward | Training Cost | Geneval |
|---|---|---|---|
| FlowGRPO | Terminal | 2.1k hours | 0.74 |
| iGRPO | Terminal | 860 hours | 0.78 |
| iGRPO | Hybrid | 490 hours | 0.80 |
| iGRPO | Instant | 205 hours | 0.82 |

*Table 5.* Reference model comparison. We compare using the EMA model versus the base model as the GRPO reference. iGRPO benefits more from a base model reference and consistently achieves higher Geneval scores than FlowGRPO under both settings.

| Method | Reference Model | Geneval |
|---|---|---|
| FlowGRPO | EMA | 0.76 |
| FlowGRPO | base model | 0.74 |
| iGRPO | EMA | 0.78 |
| iGRPO | base model | 0.82 |

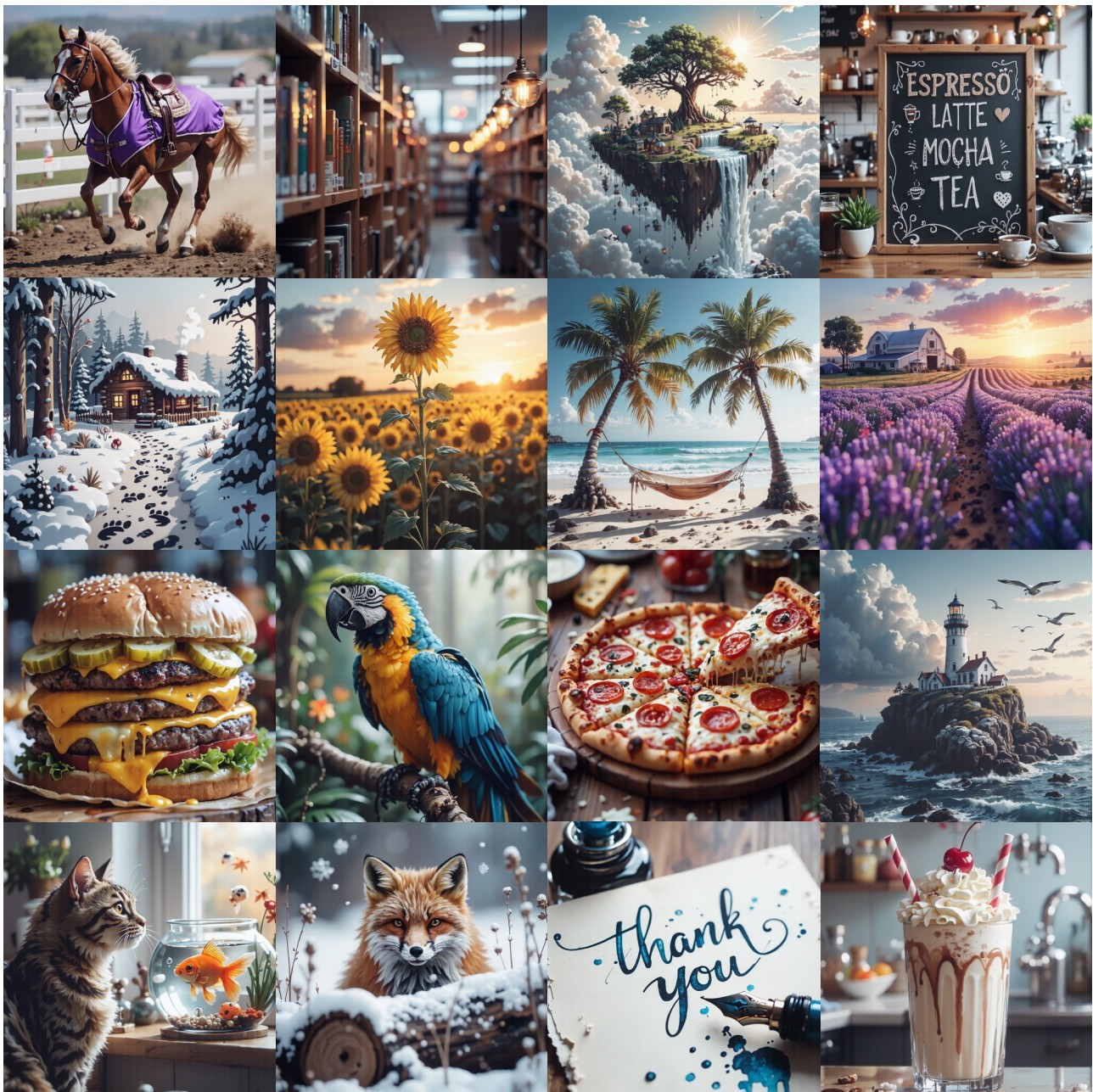

*Figure 6.* Generated samples. iGRPO produces high-quality images that align closely with both text prompts and human preferences.

the reference trajectory, the choice of the reference policy in the per-step $\text{KL}(\pi_\theta \| \pi_{\text{ref}})$ regularizer is more consequential. We compare two standard instantiations for $\pi_{\text{ref}}$: (i) an exponential moving average (EMA) teacher of the current policy and (ii) the pre-RL base model. As reported in Table 5, FlowGRPO benefits slightly from an EMA reference (0.76 vs. 0.74), consistent with its weaker per-step exploration where a tighter anchor improves stability. In contrast, iGRPO attains its best score with the base-model reference (0.82 vs. 0.78), suggesting that under stronger exploration a less restrictive reference allows beneficial policy

drift while the EMA can over-constrain updates. Unless otherwise noted, we adopt the base-model reference for iGRPO in subsequent experiments.

## 5. Conclusion

We introduced iGRPO, an online RL framework for flow-matching models that replaces full-trajectory rollouts and delayed terminal rewards with single-step updates and instant reward. Instead of traversing the entire denoising path, iGRPO maps noise (or noisy data) directly to the data space

once, then re-noises the result back to intermediate states. This decouples exploration from long SDE trajectories and turns training into a shallow, single-step procedure while still exposing the policy to a rich distribution over timesteps and noise levels. iGRPO computes rewards immediately yielding step-local signals that better reflect the heterogeneous criticality of different denoising phases. This design enables more accurate reward assignment than terminal only rewards. Our results suggest that jointly shortening rollouts and relocating rewards from terminal to instant signals provides a promising foundation for future RL for visual generation methods.

## Impact Statement

By replacing full denoising-trajectory rollouts with single-step, timestep-local rewards, iGRPO can make online RL alignment for flow-matching generators much faster and cheaper—lowering barriers to adapt high-quality generative models in education, small labs, prototyping, accessibility tools, and creative workflows while reducing energy use. However, making generators more capable and inexpensive can also amplify misuse (e.g., misinformation, impersonation/deepfakes) and entrench dataset biases, so deployments should pair efficiency gains with transparency and documentation, safety evaluation/red-teaming, provenance or watermarking when appropriate, and strong usage policies, rate limits, and content moderation.

### Acknowledgement

This work is supported by ONR N000142312641.

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
