# OpenReview forum: "iGRPO: Fast Online RL for Flow Matching Model with Instant Reward"
_ICML.cc/2026/Conference — ICML 2026 regular_

### Official Review · Reviewer_Sjz6 · 2026-03-12

**Soundness:** 2
**Presentation:** 3
**Significance:** 3
**Originality:** 2
**Overall Recommendation:** 3
**Confidence:** 4

**Summary:**

This paper introduces iGRPO, a novel online reinforcement learning framework designed for the efficient alignment of flow matching-based text-to-image generative models. Unlike traditional methods such as FlowGRPO that rely on full denoising trajectories and sparse terminal rewards, iGRPO employs a single-step mapping from noise to data and assigns instant rewards after each step. To maintain stochasticity for exploration, it incorporates a re-noising mechanism. Extensive experiments on benchmarks demonstrate that iGRPO achieves performance comparable to or better than FlowGRPO in only 1k iterations, significantly reducing training costs and improving text-image alignment.

**Compliance With Llm Reviewing Policy:**

Affirmed.

**Key Questions For Authors:**

1.Could the authors provide a detailed explanation of the re-noising mechanism, including the specific noise distribution, step size, and how these hyperparameters were chosen? If a state is re-noised multiple times, how does this differ fundamentally from full-trajectory sampling? Are there ablation studies demonstrating the necessity of re-noising and the optimal choice of its hyperparameters？
2.The paper does not report how image quality metrics (e.g., aesthetic scores, FID, LPIPS) or generation diversity evolve during training. Is there evidence that iGRPO does not suffer from reward hacking? How effective is the KL penalty in preserving diversity, and how sensitive are the results to the KL weight？
3.Is there a theoretical risk that instant rewards could bias the policy towards suboptimal solutions that are not aligned with the true objective defined by a full-trajectory ODE? Is there any guarantee of convergence or equivalence to full-trajectory optimization？

**Limitations:**

1.The implementation and hyperparameter sensitivity of the re-noising process are not disclosed, hindering reproducibility and practical application.
2. The study does not assess critical aspects like generation quality (e.g., aesthetics, perceptual similarity) and diversity, nor does it monitor for potential reward hacking or mode collapse
3.The work lacks a theoretical analysis of convergence and optimality, leaving the potential risks and biases of the instant-reward design unexplored.

**Strengths And Weaknesses:**

Strengths：
1.iGRPO's core design of single-step rollouts and instant rewards effectively addresses the inefficiency of trajectory-level credit assignment, leading to substantially faster convergence and higher sample efficiency.
2.The method is rigorously evaluated across multiple datasets and reward models , consistently outperforming strong baselines like FlowGRPO and DanceGRPO.
3.The paper provides clear empirical motivation, such as the observation of a U-shaped reward variance pattern, which aligns with the coarse-to-fine generation behavior of flow matching and justifies the use of timestep-dependent instant rewards.
Weaknesses：
1.The critical re-noising mechanism lacks detailed explanation, including its specific implementation, key hyperparameters (e.g., noise scale, step size), and ablation studies on its sensitivity
2.The paper does not analyze the trend of quality metrics (like Aesthetic Score, DeQA, ImageReward) as the reward increases, nor does it evaluate the impact on generation diversity, leaving the potential for reward hacking unaddressed
3.The role of the KL penalty term is not analyzed. There is no discussion on the sensitivity of its weight or its effect on maintaining generation diversity
4.The study fails to compare against relevant DPO-style methods like Diffusion-DPO, which weakens the positioning of iGRPO within the broader alignment landscape
5.The paper lacks theoretical guarantees on convergence or optimality. The risk that instant rewards might lead the policy to suboptimal local minima is not discussed

---

> ### Author Rebuttal · Authors · 2026-03-31
>
> **Q1: re-noising mechanism.**
>
> The re-noising mechanism is inspired by consistency models which is able to generate visually reasonable image in one step, in the sense that iGRPO avoids relying on full multi-step denoising trajectories for reward computation. Specifically, instead of traversing the entire reverse process as in FlowGRPO, iGRPO performs a one-step transition from $x_t$ predict clean state
>
> $\hat{x}_0$ = $x_t-tv(x_t, t, c)$
>
> and add noise on predicted clean state to get next stage $\hat{x}_{t-\Delta t} $ according to the policy kernel (eq 8), we can sample any steps but the reward is instant based on eq 10 and eq 11. We keep the hyperparameters the same as in FlowGRPO for a fair comparison; in particular, the maximum trajectory length is set to the same as FlowGRPO but we do not sample the whole trajectory.
>
> **Q2: reward hacking.**
>
> We would like to clarify that our current results already provide nontrivial evidence against severe reward hacking. As shown in Table 3, when training with in-domain rewards, iGRPO also improves several out-of-domain evaluators, including PickScore, ImageReward, and Unified Reward. Such consistent cross-evaluator gains demonstrate our iGRPO avoids reward hacking. In Table 1, with CLIPScore as reward, iGRPO substantially improves the subcategories  which do not rely on clip but detection model while evaluating (like Single object, Two objects, Counting, Position), showing that the gains transfer to a more generalized benchmark instead of reward hacking. We also provide the results of Aesthetic and DeQA below as review request.
>
> | Model | Aesthetic | DeQA |
> |---|---:|---:|
> | SD3.5-M | 5.39 | 4.07 |
> | FlowGRPO (8k iter) | 5.25 | 4.01 |
> | iGRPO (1.5k iter) | 5.34 | 4.09 |
>
> **Q3: KL penalty**
>
> We thank the reviewer for this comment. Our goal is not to redesign the KL term in GRPO, and thus we follow the weight of KL penalty in standard FlowGRPO. As shown in Table 5, We ablate the reference model, iGRPO performs best with the base-model reference (0.82) rather than EMA (0.78). We provide the results of KL weight =0 here which shows the importance of KL penalty when keep the image quality.
>
> | Model | GenEval | Aesthetic | DeQA |
> |---|---:|---:|---:|
> | SD3.5-M | 0.63 | 5.39 | 4.07 |
> | iGRPO (no KL) | 0.96 | 4.95 | 2.73 |
> | iGRPO | 0.96 | 5.34 | 4.09 |
>
>
> **Q4: Compared with DPO-style**
>
> Thanks for the suggestion. Diffusion-DPO achieves 0.282 HPSv2 and our iGPRO achieves 0.351. We will add this comparison in the revised version.
>
>
> **Q5 and KQ3 and L3:   instant rewards and suboptimal**
>
> Our goal here is not to guarantee the global optimum as other GRPO for flow matching model, but to develop a practical and efficient online RL method for flow-matching models. That said, our goal is more modest: to show that this surrogate provides a sufficiently informative step-local learning signal for practical online RL. In iGRPO, the instant reward is used for relative policy improvement rather than as an exact estimator of the full-trajectory objective. From this perspective, exact equivalence is not required for the method to be useful; what matters is whether the surrogate induces stable and beneficial policy updates in practice. To reduce harmful drift, iGRPO also retains an explicit per-step KL regularization to a reference policy, which constrains the updated policy from deviating arbitrarily far from the pretrained flow model. Empirically, we do not observe evidence that instant rewards drive the policy to suboptimal local minima; instead, Table 1/2/3 show clear gains in both performance and convergence, while also demonstrating strong generalization to out-of-domain evaluation metrics.
>
> **KQ1: Difference with full-trajectory sampling.**
>
> As shown in Table 4, if we re-noised multiple times and only apply the reward on the ''terminal state'', it shows similar but slight better than FlowGRPO However, in practice, we assign ''instant'' rewards on each state and do not need to sample the whole trajectory to get the "long-distance" reward. As shown in Figure 4, the reward variance keep the same at different time step while our iGPRO shows higher exploration at early timesteps and lower exploration at later timesteps, aligning with the characteristic behavior of flow-matching inference that generate structural information first and details later.
>
> **KQ2: more metrics.**
>
> | Model | Aesthetic Score ↑ | FID ↓ | LPIPS ↓ |
> |---|---:|---:|---:|
> | SD3-M | 5.39 | 27.98 | 0.75 |
> | SD3-M + FlowGRPO | 5.25 | 27.90 | 0.76 |
> | SD3-M + iGRPO | 5.34 | 27.88 | 0.75 |
>
> Thanks for the suggestion. We add the results above. These results shows that our method does not sacrifice the image quality.
>
> **L1: hyperparameter and reproducibility.**
>
> We keep the hyperparameter the same as FlowGRPO, and we will release the code and pretrained model.

---

> > ### Author Rebuttal · Reviewer_Sjz6 · 2026-04-03
> >
> > Thank you for your rebuttal. I appreciate the additional experiments and clarifications. However, several key issues from my original review remain only partially resolved.
> >
> > Q1 (Re‑noising mechanism)
> > Do you have any ablation study varying the noise scale  to show its effect on exploration vs. quality?
> >
> > Q2 (Reward hacking)
> > Could you please provide a plot or table showing Aesthetic Score and DeQA at multiple training checkpoints for iGRPO vs. FlowGRPO?
> > This would help verify that quality does not temporarily collapse due to reward hacking.
> >
> > Q3 (KL penalty)
> > How sensitive is performance to the KL weight?
> > Does the KL weight affect generation diversity beyond the single point you reported?
> >
> > Q4 (DPO‑style comparison)
> > Could you also report the GenEval overall score for Diffusion‑DPO (or a comparable DPO method) on the same setup?
> > A brief explanation of why iGRPO outperforms (e.g., online vs. offline, instant rewards) would be helpful.
> >
> > Q5 (Theoretical concern)
> > You clarified that global optimality is not guaranteed. However, the risk of suboptimal local minima remains a concern.
> > Can you provide a more intuitive explanation, grounded in your empirical observations, of why instant rewards are unlikely to trap the policy in a poor local optimum?
> > Have you observed any case where iGRPO converges to a worse solution than FlowGRPO?

---

> > > ### Author Response · Authors · 2026-04-05
> > >
> > > Q1: Re‑noising mechanism
> > >
> > > Thanks for the suggestions. Yes, increase the noise level lead to better exploration but have limited impact on image quality.  All models are trained 1500 iterations.
> > >
> > > | Model | GenEval | Aesthetic | DeQA |
> > > |---|---:|---:|---:|
> > > | SD3.5-M |0.63| 5.39 | 4.07 |
> > > | iGRPO-noise-0.1 |0.78 |5.32	  | 4.00|
> > > | iGRPO-noise-0.3 |0.85 |5.32	  | 4.05|
> > > | iGRPO-noise-0.5 |0.95 |5.35	  | 4.11|
> > > | iGRPO-noise-0.7 (default) |0.96 |5.34	  | 4.09|
> > >
> > >
> > > Q2: Reward hacking
> > >
> > > More results of Aesthetic Score and DeQA at multiple training checkpoints.
> > >
> > > | Model |  Aesthetic | DeQA |
> > > |---|---:|---:|
> > > | SD3.5-M |  5.39 | 4.07 |
> > > | FlowGRPO-500 step   |5.37 | 4.06 |
> > > | iGRPO-500 step   | 5.35	 | 4.10 |
> > >
> > > | Model |  Aesthetic | DeQA |
> > > |---|---:|---:|
> > > | SD3.5-M |  5.39 | 4.07 |
> > > | FlowGRPO-1000 step   | 5.29 |4.05  |
> > > | iGRPO-1000 step |  5.34	 | 4.07|
> > >
> > > | Model |  Aesthetic | DeQA |
> > > |---|---:|---:|
> > > | SD3.5-M | 5.39 | 4.07 |
> > > | FlowGRPO-1500 step | 5.31 | 4.01 |
> > > | iGRPO-1500 step  |5.34	  | 4.09|
> > >
> > > The aesthetic score and DeQA is very stable cross different training checkpoints. We will plot a figure and add more results to show the results in final revision.
> > >
> > > Q3: KL penalty
> > >
> > > Thanks for the suggestion. The KL penalty plays an important role. When it is removed, image quality drops significantly. Increasing the KL weight maintain the image quality but reduces exploration. However, compared with removing the KL penalty entirely, the differences among nonzero KL weights are not very significant.
> > >
> > > | Model | GenEval | Aesthetic | DeQA |
> > > |---|---:|---:|---:|
> > > | SD3.5-M | 0.63 | 5.39 | 4.07 |
> > > | iGRPO (no KL) | 0.96 | 4.95 | 2.73 |
> > > | iGRPO (kl=0.02) | 0.97 | 5.25 | 3.98 |
> > > | iGRPO (kl=0.04), default | 0.96 | 5.34 | 4.09 |
> > > | iGRPO (kl=0.06) | 0.92 | 5.35 | 4.09 |
> > >
> > >
> > >
> > > Q4: DPO‑style comparison
> > >
> > > | Method | GenEval |
> > > |---|---:|
> > > | SDXL | 0.53 |
> > > | SDXL + Diff-DPO [1] | 0.60 (+0.07) |
> > > | SD3-M | 0.63 |
> > > | SD3-M + iGRPO | 0.96 (+0.33) |
> > >
> > > iGRPO significantly outperforms Diff-DPO because Diff-DPO is still an offline preference-fitting method, whereas iGRPO is online policy optimization. Diff-DPO is closer to learning a static chosen-vs-rejected decision boundary from a fixed dataset, while iGRPO is closer to RLHF, where the model continuously optimizes against its "own" on-policy mistakes. This distinction is crucial for compositional generation, where failure modes "evolve" during training. Moreover, iGRPO provides denser, timestep-local reward assignment, whereas Diff-DPO only supervises final pairwise preference. As a result, iGRPO not only explores better than offline DPO, but also corrects generation errors more precisely and efficiently.
> > >
> > > Q5: Theoretical concern
> > >
> > > 1） if iGRPO were over-optimizing a poor local optimal, we would expect gains on the training reward to not transfer to external evaluations. However, this is not what we observe. In Tables 1 and 2, improvements with CLIPScore as reward transfer to stronger compositional performance on GenEval. In Table 3, training with in-domain rewards also improves multiple out-of-domain evaluators, including PickScore, ImageReward, and Unified Reward. Such cross-benchmark consistency is difficult to reconcile with the policy being trapped in a poor local optimal. Besides, in our current experiments, we did not observe a case where iGRPO converged to a worse final solution than FlowGRPO
> > >
> > > 2）The optimization is not unconstrained. iGRPO still uses a reference-policy KL regularization, which limits policy drift away from the pretrained model. Intuitively, this reduces the chance that a noisy instant reward at a single timestep can push the policy into a highly degenerate region of the solution space. Rather than freely chasing the surrogate reward, the policy is encouraged to improve alignment while remaining near a strong pretrained generator.
> > >
> > > 3） From an intuitive perspective, instant reward is similar to giving feedback on an intermediate draft and the final submission. Although this intermediate feedback can be noisy, it is still useful as long as it is directionally correlated with better final generations.  Besides, compared with 1000 steps in flow matching training, 40 steps inference or 10 steps in flow matching is also not global optimal. More broadly, our goal is not to recover the exact full-trajectory optimum. In practice, flow matching itself is also on approximate finite-step procedures: training is performed over 1000 sampled timesteps rather than exact continuous-time optimization, and inference typically uses only a limited number of rollout steps (e.g., 40-step inference, or 10-step rollout in FlowGRPO).

---

### Official Review · Reviewer_H4aX · 2026-03-12

**Soundness:** 3
**Presentation:** 3
**Significance:** 3
**Originality:** 3
**Overall Recommendation:** 5
**Confidence:** 2

**Summary:**

This paper considers the important problem of RL finetuning text to image flow models.

Existing methods like flow-GRPO use the terminal reward at the end of denoising as the only supervision signal,

This paper proposes IGRPO, a modification to flow-GRPO style methods, which converts the sparse, end of denoising reward to a dense, per-denoising step reward.

For computing the dense reward, they query the reward model on an "intermediate image" formed by adding the current velocity times remaining time to the current interpolant.

The method seems quite effective.

**Compliance With Llm Reviewing Policy:**

Affirmed.

**Final Justification:**

The rebuttal resolved my concerns, so I am happy to maintain the positive score.

**Key Questions For Authors:**

It could be interesting to discuss why querying the clipscore reward model on intermediate and noisy images can still give a useful reward signal. Because I would imagine the reward model itself might have been trained on only clean images?

**Limitations:**

yes

**Strengths And Weaknesses:**

Strengths.

1. The paper is very well written and easy to follow.
2. The proposed algorithm is quite simple to implement and seems to work very well.
3. The idea of using intermediate images to produce dense rewards is conceptually appealing.


Weaknesses.

1. While not a weakness as such, it could be interesting to discuss why querying the clipscore reward model on intermediate and noisy images can still give a useful reward signal. Because I would imagine the reward model itself might have been trained on only clean images?

---

> ### Author Rebuttal · Authors · 2026-03-31
>
> **Q1: CLIPScore**
>
> We thank the reviewer for this interesting question. We would like to clarify that iGRPO does not query the reward model on raw intermediate noisy latents. Instead, inspired by consistency-style models which is able to generate visually reasonable image in one step, iGRPO first performs a one-step mapping from the noisy state toward the data space, and then evaluates reward on a post-action terminal prediction obtained by collapse-and-score. Therefore, the reward is assigned to an image that is already substantially cleaner than the original noisy state, rather than to the noisy latent itself as shown in Figure 3.
> Empirically, the resulting signal is indeed useful. Figure 2 shows that CLIPScore improves much faster under iGRPO than under FlowGRPO, and Figure 4 shows nontrivial timestep-dependent reward variance rather than degenerate or uninformative scores. Moreover, optimization with CLIPScore also transfers to better GenEval performance, suggesting that the reward is not merely noisy but provides meaningful guidance for alignment.

---

> > ### Author Rebuttal · Reviewer_H4aX · 2026-04-03
> >
> > Thanks for the clarification, I am happy to maintain my positive score.

---

### Official Review · Reviewer_EBKd · 2026-03-13

**Soundness:** 3
**Presentation:** 3
**Significance:** 2
**Originality:** 3
**Overall Recommendation:** 4
**Confidence:** 4

**Summary:**

This paper focus on the GRPO improvements on image diffusion models. Formal methods such as FlowGRPO need to denoise the whole trajectory to get a clean image for reward modeling, thus it is less efficient. This paper proposes to direct predict x0 from current step to get a instant reward, which can improve the grpo efficiency as well as final performance.

**Compliance With Llm Reviewing Policy:**

Affirmed.

**Final Justification:**

Authors have address my questions in the rebuttal stage, and I will maintain my score for this paper.

**Key Questions For Authors:**

1. Can you give more analysis and results on the instant reward of high area? Does it introduce bias when high noise does not get reasonable generation results?
2. Authors should add more discussion over reward hacking.
3. Give more analysis over the denoise trajectory change when apply the one step denoise instant reward.

**Limitations:**

Yes.

**Strengths And Weaknesses:**

Strength
1. The overall motivation and methods is written very clearly, the proposed method is technically sound.
2. The results demonstrates both performance and RL efficient improvements over other methods.

Weakness
1. The instant reward may introduce  bias on the high noise area, because instant predict x0 from a high noise latent may not get reasonable result.
2. There is no reward hacking discussion. Can the instant reward make the model more vulnerable? The reward signal improve very fast, did you observe any hacking phenomenon?
3. One step reward may cause the denoise trajectory shift from original trajectory, this may cause some other side effect, hope the author can add some discussion over this topic.

---

> ### Author Rebuttal · Authors · 2026-03-31
>
> **Q1: instant predict x0 from a high noise latent may not get reasonable result.**
>
> We would like to clarify that this concern is exactly one motivation for our design. In standard FlowGRPO, intermediate high-noise states are indeed too noisy and blurry to support meaningful instant reward. In contrast, iGRPO is inspired by consistency-style models which is able to generate visually reasonable image in one step, so the reward is not assigned on the raw high-noise latent itself, but on a post-action terminal prediction obtained by collapse-and-score. This design provide a useful relative signal for step-local credit assignment. As illustrated in Fig. 3, this one-step prediction is already sufficiently clean to yield informative reward signals, enabling dense step-local supervision that is unavailable to FlowGRPO. Empirically, we do not observe evidence that this design becomes unreliable in the high-noise regime.
>
> **Q2: reward hacking**
>
> Thanks for the suggestion, we also take this into consideration. Our current results already provide nontrivial evidence against severe reward hacking. In particular, when training with in-domain rewards, iGRPO also improves several out-of-domain evaluators, including PickScore, ImageReward, and Unified Reward as shown in Table 3. Such consistent cross-evaluator gains are difficult to reconcile with pure reward hacking to a single reward model. Besides, under CLIPScore reward, iGRPO substantially improves the subcategories which relies on detection model in GenEval (like Single object, Two object, Counting, Position) in Table 1, showing that the gains transfer to a more generalized benchmark instead of reward hacking.
>
> **Q3: one step reward, Trajectory shift concern.**
>
> We thank the reviewer for this comment. Our key hypothesis is that RL for flow-matching models need not remain tightly coupled to the exact full reverse trajectory when the learning signal depends on generated data quality. Instead, iGRPO uses a one-step stochastic transition with immediate reward assignment to provide sharper credit assignment and lower sampling cost. At the same time, this shift is controlled rather than arbitrary, since the method still uses a Gaussian policy kernel together with KL regularization to a reference policy. Following GRPO, this drift is moderated by the per-step KL regularization to a reference policy, which explicitly constrains the updated policy from moving too far from the pretrained model. Empirically, we do not observe harmful effects from this trajectory deviation; on the contrary, it leads to faster convergence and better final alignment.

---

> > ### Author Rebuttal · Reviewer_EBKd · 2026-04-02
> >
> > The authors have address my questions. I have no further issues about this paper.

---

> > > ### Author Response · Authors · 2026-04-02
> > >
> > > Thank you again for the constructive feedback!

---

### Official Review · Reviewer_j65g · 2026-03-23

**Soundness:** 3
**Presentation:** 3
**Significance:** 2
**Originality:** 2
**Overall Recommendation:** 4
**Confidence:** 4

**Summary:**

This paper modifies the GRPO algorithm in the context of flow matching models to use an estimate of the reward at each step of the flow rollout by looking ahead with the velocity prediction. It then benchmarks the speed of convergence it achieves compared to flowGRPO is superior.

**Compliance With Llm Reviewing Policy:**

Affirmed.

**Key Questions For Authors:**

n/a

**Strengths And Weaknesses:**

Soundness: The paper makes a smart observation that rewards should be evaluated throughout the whole rollout. However, it seems insufficient to use the velocity prediction at each step as a surrogate for the flow map to try and evaluate the lookahead reward. E.g. equation 10 kind of buries this under the rug, in the instant reward section. In particular, at times close to the noise, the velocity lookahead gives almost totally meaningless signal to the reward and the adaptation to the trajectory would be extremely noisy. There have been a number of recent works [1,2,3] that have shown that actually using the one-step map to look ahead is much more efficient than the denoiser or velocity prediction. The reviewer is curious whether using the velocity prediction is inhibiting the method. More importantly, it is swept under the rug that the velocity prediction is not actually correct for this setting. This should be made clear.

Presentation: It reads fine, though it's a bit weird to write everything like the flow starts at some arbitrary time $t$.

Significance: Scaling up GRPO would be useful for flows. The results are a bit lukewarm in the sense that the numbers generally are not outside a standard deviation from flowGRPO, though they did converge faster.

Originality: This work fits into how  many papers are thinking about properly evaluating reward models throughout trajectories. Applying it to GRPO is novel, as I understand it.




1. "Test-time scaling of diffusions with flow maps" Amirmojtaba Sabour, Michael S. Albergo, Carles Domingo-Enrich, Nicholas M. Boffi, Sanja Fidler, Karsten Kreis, Eric Vanden-Eijnden, arXiv:2511.22688
2. "Meta Flow Maps enable scalable reward alignment" Peter Potaptchik, Adhi Saravanan, Abbas Mammadov, Alvaro Prat, Michael S. Albergo, Yee Whye Teh, arXiv:2601.14430
3. "Diamond Maps: Efficient Reward Alignment via Stochastic Flow Maps" Peter Holderrieth, Douglas Chen, Luca Eyring, Ishin Shah, Giri Anantharaman, Yutong He, Zeynep Akata, Tommi Jaakkola, Nicholas Matthew Boffi, Max Simchowitz, arXiv:2602.05993

---

> ### Author Rebuttal · Authors · 2026-03-31
>
> **Q1: velocity prediction.**
>
> We thank the reviewer for this comment. Our current formulation uses one step transition as a practical collapse surrogate for immediate reward assignment, rather than claiming to reconstruct the exact terminal sample. Our design is inspired by consistency model which is able to generate visually reasonable image in one step and the consistency model can be easily finetuned from a flow matching model. Therefore, eq.10, the one-step predictions (from purely noise to clean state) can preserve enough semantic content to support downstream evaluation. We will add more visualizations on high noise state to clean state prediction. Our claim is thus not that the velocity-based collapse is theoretically exact, but that it provides a sufficiently informative step-local look ahead signal for online RL. In iGRPO, this reward is used for relative advantage estimation, so it need not be a perfect estimator of the final rollout reward to be useful for policy improvement. In our Eq. (5), the velocity parameterization naturally induces a implicit clean-state prediction $\hat{x}_0= x_t-tv(x_t, t, c)$. Empirically, the strong convergence improvements in Fig. 2 and the timestep-dependent reward behavior in Fig. 4 further suggest that the one-step signal is informative rather than dominated by noise.  We also appreciate the reviewer pointing us to learned one-step flow maps [1,2,3]. iGRPO does not assume access to a separately trained one-step flow map; instead, it works directly with the existing velocity-parameterized flow-matching model and uses the induced clean-state reconstruction as a lightweight surrogate for immediate reward assignment. we will cite and discuss these papers in the revision.
>
>
>
> **Q2: Significance**
>
> We believe the results represent a clear practical gain when considered jointly across performance, generalization, and efficiency. In Table 1, iGRPO improves overall GenEval from 0.74 to 0.82 while using only 10\% of the training iterations of FlowGRPO under CLIPScore reward. In Table 2, the absolute gap is smaller (0.96 vs. 0.95 under GenEval reward), but this is also a near-saturated regime where both methods already approach the benchmark ceiling; importantly, iGRPO still attains the better score with substantially fewer iterations (1.5k vs. 8k). Beyond GenEval, Table 3 shows consistent improvements across all five metrics. Most importantly, Table 4 shows iGRPO reduces wall-clock training cost from 2,100 GPU-hours to 205 GPU-hours (10.2x reduction). We view the significance of iGRPO as lying in this joint improvement: better alignment quality at a fraction of the compute.

---

> > ### Author Rebuttal · Reviewer_j65g · 2026-04-04
> >
> > Thanks. Yeah, i'm not saying it's useless I'm just saying to be honest. This sounds like crazy marketing: "practical collapse surrogate for immediate reward assignment". Just say you are using the denoiser so you don't have a true possible outcome, but a noisy estimate, and when the noise is high, the estimate is less useful. It's not bad to say this, it doesn't make the algorithm bad. It just confuses the reader when one is not straightforward.
> >
> > If you make this clearer I can raise my score.

---

> > > ### Author Response · Authors · 2026-04-04
> > >
> > > Thank you for the clarification! Yes, it is noisy estimate when noise is high and such noisy estimate is enough in our iGRPO. Our key point is also that the estimate does not need to match the final generated image in full quality. It only needs to be visually reasonable enough for the reward model to extract useful/meaningful reward and perform relative ranking within a group. That's practical collapse surrogate. As shown in Fig. 3, although these one-step predictions are not as sharp as the final samples, they already preserve enough coarse structure and semantics for a model such as CLIP to distinguish better versus worse candidates. In other words, our goal is not exact reconstruction of the final outcome from high noise part, but a sufficiently informative state.
> > >
> > > As for why a one-step prediction from a high-noise state can still be visually reasonable, this is supported by prior work on one-step generative modeling[1,2], such as consistency models[1], which show that mapping from high noise (even pure noise) to a clean prediction is very easy in practice by finetuning a flow matching model. This is also particularly natural in our setting because flow matching already predicts the velocity field, and the corresponding clean-state estimate can be written directly as
> > > $\hat{x}0=x_t-tv(x_t, t, c) $
> > > Thus, our instant reward uses this denoised estimate as a practical look-ahead signal, rather than claiming it is an exact final outcome.
> > >
> > > [1] Song, Y., Dhariwal, P., Chen, M., & Sutskever, I. (2023). Consistency models.
> > >
> > > [2] Yin, T., Gharbi, M., Zhang, R., Shechtman, E., Durand, F., Freeman, W. T., & Park, T. (2024). One-step diffusion with distribution matching distillation. In Proceedings of the IEEE/CVF conference on computer vision and pattern recognition (pp. 6613-6623).

---

### Decision · Program_Chairs · 2026-04-30

**Decision:**

Accept (regular)

**Comment:**

The reviewers agreed that the proposed iGRPO method provides a conceptually appealing and highly efficient approach to online RL for flow matching models by leveraging step-local instant rewards. While reviewers initially raised valid concerns regarding the reliability of one-step reward surrogates at high noise levels, potential reward hacking, and missing baseline comparisons, the authors provided a thorough rebuttal with additional empirical evidence that adequately addressed these issues. The paper is technically sound, demonstrates significant training speedups, and offers a valuable contribution to the community. Therefore, the paper is recommended for acceptance.